# Founder transformants of cotton (*Gossypium hirsutum* L.) obtained through the introduction of DS-Red, Rec, Rep and CRISPR/Cas9 expressing constructs for developing base lines of recombinase mediated gene stacking

**Sabin Aslam**[1]*, **Sultan Habibullah Khan**[1,2], **Aftab Ahmad**[2,3], **Sriema Lalani Walawage**[4], **Abhaya M. Dandekar**[4]

**1** Center of Agricultural Biochemistry and Biotechnology (CABB), University of Agriculture, Faisalabad, Pakistan, **2** Center of Advanced Studies in Agriculture and Food Security (CAS-AFS), University of Agriculture, Faisalabad, Pakistan, **3** Department of Biochemistry, Faculty of Sciences, University of Agriculture, Faisalabad, Pakistan, **4** Department of Plant Sciences, School of Biological Sciences, University of California, Davis, California, United States of America

* sabinaslam@gmail.com, sabaslam@ucdavis.edu

## Abstract

Cotton being the major fiber crop across the world is exposed to numerous biotic and abiotic stresses. Genetic transformation of cotton is vital to meet the world's food, feed and fiber demands. Genetic manipulation by randomly transferring the genes emanate variable gene expression. Targeted gene insertion by latest genome editing tools results in predictable expression of genes at a specified location. Gene stacking technology emerged as an adaptive strategy to combat biotic and abiotic stresses by integrating 2–3 genes simultaneously and at a specific site to avoid variable gene expression at diverse locations. This study explains the development of cotton's founder transformants to be used as a base line for multiple gene stacking projects. We introduced Cre and PhiC31 mediated recombination sites to specify the locus of incoming genes. CRISPR-Cas9 gene was integrated for developing CRISPR based founder lines of cotton. Cas9 gene along with gRNA was integrated to target Rep (replication) region of cotton leaf curl virus. Replication region of virus was specifically targeted to diminish further proliferation and preventing the virus to develop new strains. To successfully develop these primary transformants, a model transformation system has been optimized with the red color visualization (DS-Red). Following red color transformation system, three baselines with recombination specified site (Rec), targeted replication region (Rep) and Cas9 founder lines have been developed. These founder transformants are useful for developing recombinase mediated and CRISPR/Cas9 based originator lines of cotton. Moreover, these transformants will set up a base system for all the recombinase mediated gene stacking projects.

**Data Availability Statement:** All relevant data are within the manuscript and its Supporting information files.

**Funding:** PhD research fellowship awarded to Sabin Aslam by USAID-CAS at UC-DAVIS.

**Competing interests:** No conflict of interest.

## Introduction

Cotton is the most widely produced cash crop throughout the world, comprising approximately 40% of the world's agricultural production [1]. Globally cotton is the fifth largest oilseed crop after soybean and canola [2] and is cultivated in nearly sixty countries. Global cotton production is estimated at 119.4 million bales in 2019/20 with a 3.5% decline from the previous year [3]. *Gossypium hirsutum*, American upland cotton (2n = 4x = 52) accounts for 31.66 mha (90%) of the world's cotton cultivable area [1].

The yield and quality of cultivated cotton is affected by numerous biological and abiological stressors. One such stressor is a single stranded DNA virus of family *Geminiviridae* and genus *Begomovirus* [4]. This virus originated in Nigeria in 1912 [5], which later spread to the Indian subcontinent in the twentieth century and severely affected Pakistan's cotton yield in 1967 [6]. In 1992, further wide scale outbreaks caused significant output losses of up to 0.543 m bales [7], which worsened ever since [8]. The cotton leaf curl virus (CLCuV) is transmitted by white fly (*Bemisia tabaci*), which is responsible for circulative and persistent transmission of virus [9, 10]. Multiple strains of CLCuV are currently prevailing in Africa and Asia such as India and Pakistan [11, 12]. In India, these strains are prevalent in the states of Haryana, Punjab, Uttar Pradesh and Rajasthan [13–16]. The affected African countries include Cameroon, Chad, Cote d'Ivoire, Niger, Nigeria, Sudan and Togo [17–23]. White fly not only reduces crop yield but also affects plant growth and fiber quality [24].

Primarily, Cotton leaf curl disease (CLCuD) resistant exotic germplasm was identified as LRA-5166, CP-15/2 and Cedix [25]. These varieties varied in their degree of susceptibility and were not completely resistant to CLCuV. Environmental factors like temperature and humidity were also the contributing factors towards incidence of the disease [26]. Conventional diagnostic procedures i.e; field evaluation, whitefly transmission and evaluation through grafting, could not detect the specific resistant genes within the available wild resistant germplasm [25]. Genetic modification at molecular level worked best to impart disease resistance [27–30]. Targeting virus at a specific site combat the disease but it remained adhered to impart resistance to a specific strain. For the eradication of disease, there is a dire need to integrate a multilayered defense system to avoid the enormous losses caused by CLCuD.

Gene stacking emerged as a new approach to tackle multiple problems within a single crop. It refers to the process of pyramiding two or more genes within organism's genome either at random position or at a specific site. Three strategies of stacking multiple genes in crop plants have been reported to date. First is "do over strategy" to package multiple genes into one transformation cassette and then integrating into plant. This strategy requires excessive screening to search for the plant integrated with all transgenes at a precise location [31]. Second strategy involves the direct transformation into pre-existing elite variety to stack the new trait. This approach may require frequent independent transformation trials to settle the incoming gene to a precise location [31]. Third and the most promising strategy, called the site-specific gene addition, is used to integrate new genes into previously engineered genomic location. This strategy offers introgression of site-specific gene locus into new cultivars to simplify the upcoming breeding work [31–35]. Site specific gene stacking refers to gene integration and deletion events by providing an extra recombination site to serve as a target site for the next round of DNA integration event [36]. We intend to use the third strategy for developing site specifically stacked Coker-312 for optimizing a model system. Target line was marked with specific sites to integrate gene stacks at the target location. Target line has the additional recombination sites to remove the entire cassette after the site-specific gene stacking system need to be replaced by a new one in the long run. The developed system will be introgressed into commercial cultivars by utilizing breeding procedures to combat CLCuV.

Several transformation systems for cotton are reported that are simple, effective, variety-independent and adaptable to regeneration. Two well-known systems used for transferring foreign DNA into plant genome are particle bombardment and *Agrobacterium*-mediated transformation. The first successful cotton transformation system used cotyledons of Coker-201 [37] and somatic embryogenesis of hypocotyls [38]. Later on; apical meristems, cotyledons, embryogenic suspension cells, mature embryos, shoot apex, somatic and embryonic calli were used as explants for gradual improvement of cotton transformation [29, 30, 39–57]. The major constraint in cotton transformation has been the recalcitrance of cotton varieties to tissue culture [58]. Coker cultivars have regeneration capacity, with low efficiency to develop somatic embryos.

Genome editing by using CRISPR (Clustered Regularly Interspersed Short Palindromic Repeats) system has become an efficient approach to add, delete or modify the genomes with great precision. Therefore, researchers need some base plants with integrated Cas9 gene to ease the process of initiating experiments without developing the Cas9 plants from scratch. On optimizing and visualizing each step of transformation by DS-Red, we transformed Coker-312 with three constructs separately and developed a foundation system for gene stacking in cotton. We produced red fluorescent plants to make an easy screening for separating the putative transgenic plants. Next, we developed cotton founder plants for site-specific recombination by transforming pG-Rec vector. The pG-Rec vector carried Rec cassette for recombinase-based DNA recognition sites. The efficacy of developing founder plants for other selectable markers like hygromycin was also tested. The pHSE-401-Rep was transformed carrying Cas9 gene along with gRNA against Rep region of CLCuV. A Cas9 founder vector (pCas9-Rec) carrying Cas9 gene along with recombinase mediated DNA recognition sites was transformed separately. This vector carried DNA recombination sites for site specific integration of gRNAs. The well-known *Agrobacterium* mediated transformation system offered an easy tissue culture practice to regenerate mature and fertile plants by exploiting the callus induction potential of hypocotyls derived from mature seeds. Resulting plants will serve as founder plants for multiple projects. That is, Rec lines will be used as founder lines for the gene stacking project and Cas9 founder lines will be used in CRISPR projects for targeting multiple gRNAs transiently/stably.

## Materials and methods

### Plasmid design

To examine the expression of integrated genes, we used red fluorescent protein (DS-Red), which is a powerful tool for identifying protein localization at a subcellular level. The emission of red fluorescence with longer wavelength mitigates the problem associated with light scattering and cells' self-fluorescence [59, 60].

Four different plasmids (pKGW-RR, pHSE-401-Rep, pG-Rec and pCas-Rec) were used for transformation in Coker-312. The primary event was performed with construct pKGW-RR carrying DS-Red as a reporter gene [61] (Fig 1a).

Second transformation event was performed with construct pHSE-401-Rep obtained from Addgene (Fig 1b). pHSE-401-Rep vector was transformed for combating CLCuKV and to develop a strategy for targeting the viral genome by CRISPR/Cas9 system.

Third event was done by transforming pG-Rec [62] in cotton hypocotyls for initiating the project of gene stacking. pG-Rec vector was constructed to integrate Rec cassette from pJet-Rec to pGreen-0029 by restriction digestion. Resultant vector was the Rec cassette integrated into pGreen-0029 backbone. Whole plasmid was constructed by restriction digestion/ligation (Fig 1c). The pG-Rec vector was transformed to tag the specific site within the plant genome

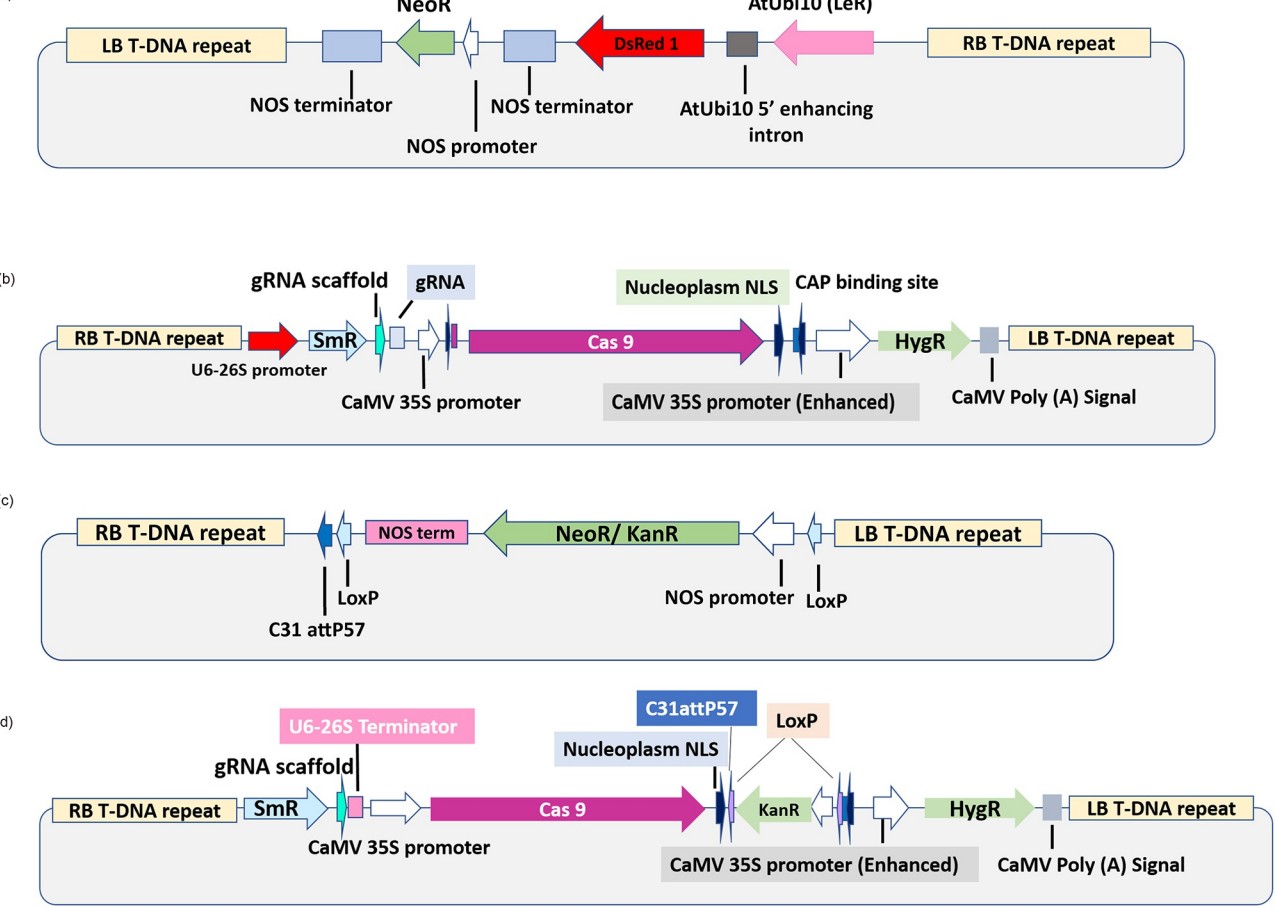

**Fig 1.** (a) pKGW-RR. DS-Red gene in pKGW-RR is transcribed by AtUbi10 (Ler) promoter and Nos terminator for plant expression. pKGW-RR was isolated from *E. coli* cells and digested for confirmation before *Agro*-transformation. pKGW-RR vector was used for optimizing Coker-312 transformation protocol for visualizing each step of transformation procedure. (b) pHSE-401-Rep. pHSE-401-Rep carried Cas9 gene (transcribed by CaMV 35S promoter) and a gRNA to target conservative region of Rep gene for Cotton Leaf Curl Khokran Virus (CLCuKV). This construct carried hygromycin antibiotic resistance gene for plant selection and kanamycin (NptII) antibiotic resistance gene for bacterial selection. (c) pG-Rec. Rec-cassette as NptII gene flanked between two directly oriented Cre mediated LoxP sites and a PhiC31 recombinase mediated attP57 site. (d) pCas-Rec. Whole Rec cassette combined with Cas9 gene to specify the locus of Cas 9 to be integrated.

for gene stacking project. The tagged site will be used as a target site for pyramiding genes of interest (GOI).

The pCas-Rec was transformed in cotton hypocotyls for developing Cas9 based founder plants. This vector was constructed by In-fusion cloning of Rec cassette within pHSE-401 restriction digestion site EcoRI. This vector carried complete Rec cassette for specifying target site by integrating PhiC31 mediated attP57 site within plant genome. LoxP sites were retained within the pCas-Rec vector for Cre recombinase mediated marker gene removal strategy. The vector also carried Cas9 gene for targeting specific gRNA (not included in this vector). Due to the presence of Cas9 gene, the vector could serve as a founder line for developing Cas9 plants. Specifying target site within the plant genome is an added advantage of plant transformation by this vector. This vector will be able to develop Cas9 based site-specific target plants with the capacity to remove marker gene by Cre recombinase mediated deletion strategy (Fig 1d).

All the constructs were transformed in *Agrobacterium* (pKGW-RR and pG-Rec in GV3-101 strain; pHSE-401-Rep and pCas-Rec in EHA-105-PCH-32 strain) for plant transformation.

## Initiation and generation of explant

The sunshine mix soil misted with nutrient water was used for Coker-312 seeds germination in dark at 28˚C (Fig 2). Seeds were initially watered two days after sowing and then on alternate days. Seeds started germinating within three to four days displaying the emergence of first two cotyledonary leaves above soil. Hypocotyls started increasing in length reaching ~10-12cm in height and were shifted to light for 48 hours to initiate the process of photosynthesis. It took two weeks to germinate the seeds, to cut the hypocotyls and to prepare them for transformation.

After 48 hours growth in light, cotton hypocotyls were detached from the root portion in soil. Cotyledonary leaves were removed with a sterile sharp surgical blade (Carbon steel scalpel blade # 10). Hypocotyls were trimmed to ~ 8-10cm with same sharp sterile blade, making blunt ends. Cut hypocotyls were immersed in 30 mL sterile distilled milli-Q water. Hypocotyls were shaken well to remove all dust particles, then rinsed with sterile milli-Q water 2–3 times. Hypocotyls were surface-sterilized with 45 mL sterilization solution (0.525% Chlorox bleach and 5 µL tween-20 in 100 mL milli-Q water), and placed on a rotating plate for 15 min at room temperature. Subsequent steps for hypocotyls handling were performed in Laminar Flow Cabinet (LFC).

The sterilization solution was discarded and the hypocotyls were rinsed with sterile distilled milli-Q water 4–5 times to wash out all the sterilization solution adhered to the surface of hypocotyls. Hypocotyls were cut with a sharp sterile surgical blade (Carbon steel scalpel blade # 11) to avoid damaging the cut ends. A centimeter portion from each end was discarded and the remaining hypocotyl was divided into 2–3 small (3-4cm) segments as explants. Both ends of the cut hypocotyl segments should be kept sharp and blunt-ended, as rough and broken ends will not be able to take up *Agrobacterium* infection properly. The hypocotyl segments were accumulated in a separate dry and clean petri plate.

## Genetic transformation

Cotton hypocotyls were infected with *Agrobacterium* culture in a non-specific Medium (5 g/L Tryptone, 5 g/L NaCl, 0.1 g/L MgSO4, 0.25 g/L KH2PO4 and 1 g/L Glycine and pH = 5.8) immediately after cutting. 25 mL *Agrobacterium* culture was poured into the petri plate of freshly cut hypocotyls. Hypocotyls were infected for 45 min without shaking. The liquid culture was pipetted out of the petri plate after 45 minutes. Hypocotyl segments were surface-dried with double sterile filter paper to wipe off as much *Agrobacterium* culture as possible. The drying step was repeated twice and hypocotyls were air-dried for 2–3 min in LFC.

Surface dried 10–12 hypocotyls were aligned on non-selective basal Incubation Medium (IM) (Regular Murashige and Skoog minimal organics medium with MS vitamin stock, 3%

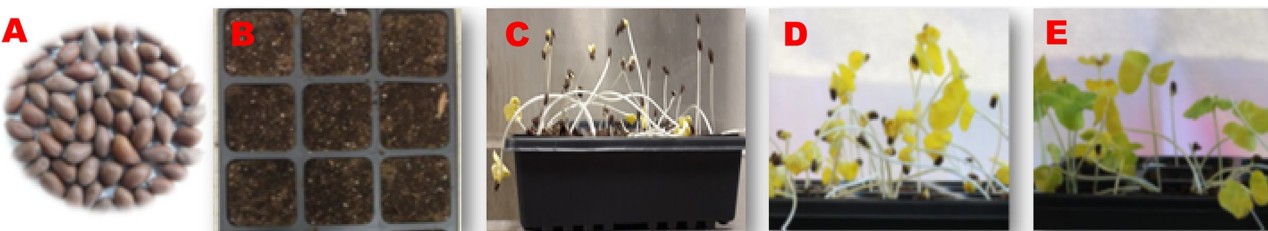

**Fig 2. Explant preparation.** (A) Coker-312 delinted seeds, (B) Seeds sown in moist sunshine mix soil in darkness, (C) Emergence of cotyledonary leaves and whitish hypocotyls, (D) Hypocotyls growth in dark (yellowish appearance), (E) Hypocotyls turned green when subjected to sunlight for 48 hours.

(w/v) Glucose, 0.4 μM 2,4-D, 0.4μM. Kinetin, 200 μM Acetosyringone, 1000 mg/L MgCl2, and 0.2% (w/v) Gelzan, pH 5.2) without embedding in that medium. The plates were placed in the dark for 48 hours at 21°C.

## Callus induction

Hypocotyls were shifted to Coker Callus Inducing Medium (CCIM) (4.33 g Murashige and Skoog minimal organics medium, 1 mL/L MS vitamin stock, 3% (w/v) Glucose, 0.4 μM 2,4-D, 0.4 μM Kinetin, 1000 mg/L $MgCl_2$, 50 mg/L kanamycin, 400 mg/L Cefotexime, and 0.2% (w/v) Gelzan, pH 5.8) with Kanamycin (pKGW-RR, pG-Rec and pCas-Rec) or Hygromycin (pHSE-401-Rep). Infected hypocotyls were retained on same CCIM medium at 28°C with 16/8 hours photoperiod. Hypocotyls were shifted to fresh medium of same formulation after 10 days. They were re-shifted on fresh CCIM medium after every two weeks, provided that all hypocotyls were clean and free from bacterial or fungal contamination. Hypocotyls started developing callus along both cut ends and attained the shape of a dumbbell, with fully grown callus on both ends of each segment after 2–3 months. Calli were separated from hypocotyls with fine sterile scalpel (Carbon steel scalpel blades # 11) and shifted to fresh selection plates of CCIM. Calli were grown on CCIM for 21 days at same temperature and photoperiod, that is 28°C and 16/8 hours respectively.

## Embryogenesis

When calli grew to cover the whole petri plate and started turning green, they were moved to dark to control photosynthesis and enhance callus growth. Calli were shifted to Coker Regeneration Medium-I (CRM-I) (4.33 g Murashige and Skoog minimal organics medium, 1mL/L MS vitamin stock, 3% Glucose, 1000 mg/L $MgCl_2$, 0.22% (w/v) $KNO_3$, 50 mg/L Kanamycin (pKGW-RR, pG-Rec and pCas-Rec) or Hygromycin (pHSE-401-Rep), and 0.8% (w/v) Phytoagar, pH 5.8). Calli were placed at 28°C with 16/8 hours photoperiod and maintained on medium of same formulation for 21 days. Cotton calli were shifted to Coker Regeneration Medium-II (CRM-II) (4.33 g Murashige and Skoog minimal organics medium, 1 mL/L MS vitamin stock, 3% Glucose, 1000mg/L $MgCl_2$, 1,000 mg/L Glutamine, 500 mg/L Asparagine, 1.25 mg/L $CuSO_4$, 100 mg/L Ascorbic acid, 2,000 mg/L Activated charcoal, 50 mg/L Kanamycin (pKGW-RR, pG-Rec and pCas-Rec) and Hygromycin (pHSE-401-Rep), and 0.8% (w/v) Phytoagar, pH 5.6–5.8) in the dark for 21 days. Upon regeneration, embryos were separated from the callus and shifted to fresh medium of same formulation with respective selective agent till the emergence of cotyledonary leaves. Coker Root Inducing Medium-I (CRIM-I) without hormones (2.215 g ½ Murashige and Skoog minimal organics medium, 1 mL/L Vitamin stock, 3% Glucose, 400 mg/L Carbenicillin, 150 mg/L Timentin (as growth enhancer), 50 mg/L Kanamycin, and 0.8% Phytoagar, pH 5.6–5.8) was used in jars for root development. Plantlets were shifted to Coker Root Inducing Medium-II (CRIM-II) (DKW medium 5.220 g/ L, 1 mL/L Vitamin stock, 3% Sucrose, 1,000 mg/L Glutamine, 500 mg/L Asparagine, 100 mg/L Ascorbic acid, 1 mL/500 mL STS buffer* and 0.8% Phytoagar, pH 5.6–5.8) (*STS Buffer: 12 mM Sodium thiosulphate + 12 mM silver nitrate) after 10 days growth in CRIM-I for root development.

Callus developed from a single side of each hypocotyl was kept separate as an independent transformation event. Two callus plates per transformation event were maintained for embryo regeneration. Embryos developed from calli originated from same transformation event were considered as alike. The plantlets appearing from these embryos were tagged as developed from same transformation event.

## Hardening

When 5–6 leaves grew on CRIM-II, the embryos started to develop a complete plant with enough leaves for photosynthesis. Hardening was initiated by opening jars briefly during the day. The frequency of opening jars was increased to every following day according to the health condition of the plant. Jars were kept open for 24 hours after 4–5 days. Plants were shifted to sunshine mix in pots and were placed at 28°C. They were covered with a dome to allow them to gradually adapt to an open-air environment. Hardening process took 2–3 weeks to completely acclimatize the plants to an open-air environment. Hardened plants were then shifted to the greenhouse until full maturity. Cotton flowers were bagged and tagged before opening to avoid cross-pollination with pollens from the surrounding plants. Bags were removed upon emergence of cotton bolls and plants were kept under observation until seeding. Cotton was picked at maturity and seeds were harvested and delinted (Fig 3).

## Transgene analysis

Genomic DNA was extracted from plants transformed with all the four plasmids: pKGW-RR, pG-Rec, pHSE-401-Rep and pCas-Rec. DNA was extracted at the callus and leaf stage by taking 200 mg callus/leaf from each plate displaying a well-defined single transformation event. Callus separated from either side of a single hypocotyl was marked as an independent transformation event. Four callus plates from independent transformation events were randomly selected for DNA extraction. DNA extraction was done using the QIAGEN DNeasy plant mini kit. PCR was performed using six different pairs of primers to screen positively transformed plants for all vectors separately. The names of oligos used for each separate transformation event, their sequences and the gene to be amplified for transformation confirmation are shown in Table 1.

## Southern blotting

The identification of DS-Red, Npt-II and Cas9 gene integrated within genome of cotton was inveterated by southern blotting. The DNA of PCR positive samples was lyophilized (Christ Freeze Dried) after extracting about 300–500µg DNA per sample. Vacuum generation and absolute drying were done for ~6 hours at -86°C and 1mbar. Lastly, dried DNA samples were re-suspended into sterile milli-Q-water with a final concentration of ~5 µg/ µL. These samples were subjected to restricted digestion (100 µg per DNA sample, 1U EcoRI-HF, 10X Cutsmart buffer) at 37°C for 2 hours. Whole reaction mixture was gel electrophoresed at 0.8% agarose gel. The 1kb ladder was used as DNA marker to compare fragment size. Gel was run at a low voltage (~40V) for 7–8 hours to release fine fragments of DNA. Fragments were visualized by gel documentation system.

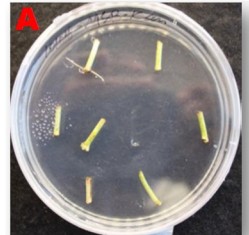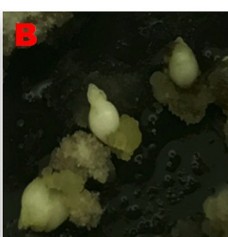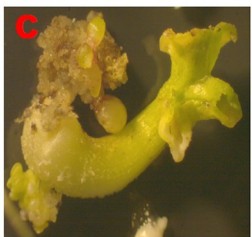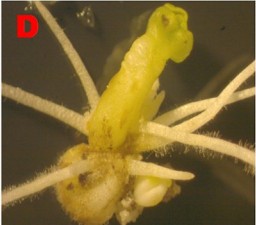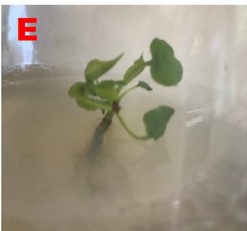

**Fig 3. Cotton transformation stages.** (A) *Agro*-transformed hypocotyls, (B) Embryos regenerated from callus, (C) Shoot development, (D) Root development, (E) In-Vitro plant.

**Table 1. Primers used for transgene analysis.**

| Name of Oligos | 5′→3′ Sequence | Purpose |
|---|---|---|
| NptII For. | AGTCCCGCTCAGAACAACTC | NptII gene amplification 557bp |
| NptII Rev. | GTTGTCACTGAAGCGGGAAG | |
| SbfI-AtDS-Red For. | CCTGCAGGGAATTCGCCCTTGGGCCC | DS-Red amplification 2489bp |
| NsiI-At-DS-Red Rev. | ATGCATGTGGTTGGTGCTTTCCTTACATTCT | |
| M-13 For. | GTAAAACGACGGCCAGT | Construct pHSE-401 7789bp |
| M-13 Rev. | CAGGAAACAGCTATGAC | |
| Cas9 For. | GACAAGAAGTACTCGATCGGCCTC | Cas9 amplification 4266bp |
| Cas9 Rev. | GTCGCCCCCGAGCTG | |
| Rec For. | GACCCTACGCCCCCAACT | Rec cassette confirmation 1493bp |
| Rec Rev. | TCGTCGTCGACAAGCCGGATAA | |
| pG.Rec-pHSE-401-IF For. | gattgacaacgaattCATAACTTCGTATAATGTATGCTATACGAAGTTATTCTAGATTTATCC | IF primers for pCas-Rec construct confirmation 1449bp |
| pG.Rec-pHSE-401-IF Rev. | ccatgattacgaattATAACTTCGTATAGCATACATTATACGAAGTTATTCTAGATGAG | |

Gel was depurinized by shaking for 15 minutes with 0.25M HCl. DNA was then prepared for blotting by treating the gel with denaturation buffer (0.5M NaOH + 1.5M NaCl) and shaken well at orbital shaker for 15 minutes. Next, Gel was treated with neutralization buffer (1M Tris-HCl and 1.5M NaCl) two times for 15 minutes in an orbital shaker, and then placed on a clean table. A piece of nylon membrane, immersed in neutralization solution, was placed on the gel avoiding any air bubble. Three pieces of whatmann filter paper were soaked in neutralization solution and then put on the gel. A stack of paper towels and a weight of about 250g was put at the top. Gel was blotted for 4–5 hours. Paper towels were then removed and gel was labelled accordingly. Nylon membrane was placed in UV-Stratalink and auto-crosslink was pushed to covalently bind DNA with the membrane. Membrane was then stored at 4˚C for few minutes before hybridization.

Membrane was pre-hybridized by rolling in a hybridization tube by facing membrane edge in a rotating direction. Incubation was done for an hour at 60˚C in pre-hybridization buffer (5X SSC, 2% W/V Blocking reagent, 0.1% N-laurylsarcosine, 0.02% W/V SDS). Probe was denatured by heating at 95˚C in microcentrifuge for 10 minutes. Membrane was then incubated in hybridization buffer (10 mL pre-hybridization buffer + 50 ng probe DNA) overnight at 63–65˚C.

Membrane was washed for 15 minutes at a room temperature in 2X SSC (15mL) and 0.1% SDS. It was washed twice for 15 minutes at 55˚C in 0.5X SSC and 0.1% SDS and then washed with wash buffer (0.3% W/V Tween-20 in maleic acid) 3–4 times at a room temperature. Incubation was done in blocking buffer (5% W/V Milk powder in maleic acid) for 30 minutes. Membrane was incubated in Anti-DIG-AP conjugate solution for 30 minutes and then washed three times with wash buffer (100 mL) for 15 minutes. Buffer-3 (0.1M NaCl, 0.1M Tris-Cl and 50mM $MgCl_2$) was used to equilibrate for 3–5 minutes. Washed membrane was taken out in a flat bowl and was incubated with gentle shaking for 5–10 minutes. Wet membrane was sealed in hybridization bag and was exposed to X-ray film for 15 seconds—5 minutes to develop film. Probe was removed and membrane was free to use for hybridization with a different probe. Membrane then was incubated for 30 minutes at 45˚C in 0.4M NaOH (100mL) and washed with 1X SSC and 0.1% SDS at room temperature.

## Results

Our study emphasized on the development of founder plants carrying target cassettes (pKGW-RR, pG-Rec, pHSE-401-Rep and pCas-Rec) that ultimately led to the development of

target lines in cotton. The positive transgenic plants were examined at multiple stages of growth pattern, from callus induction after stable transformation till regeneration of callus to embryos and then to mature plants. Integration of Rec, Rep and Cas9 cassettes within these plants initiated multiple experiments for future use (gene stacking and CRISPR/Cas9 targeting). The study mainly focuses on developing Rec and Cas9 positive transgenic plants to initiate CRISPR/Cas9 based gene stacking projects.

Seeds germinated to whitish hypocotyls after 2 weeks of growth in soil mixed with nutrient water. On exposure to sunlight for 48 hours, they turned green, manifesting the initiation of photosynthesis by cotyledonary leaves. Light absorbance and food production rendered the hypocotyls green and resilient towards *Agrobacterium* infection. After transformation of hypocotyls with all the three plasmids (pG-Rec, pHSE-401–Rep and pCas-Rec), callus growth was observed 14 days after transforming the hypocotyls on respective antibiotic plates (Kanamycin for pG-Rec and pCas-Rec; Hygromycin for pHSE-401). Somatic calli were pale yellow in color. Callus turned greenish after 21 days growth under 16/8 hours photoperiod. Callus was unable to undergo photosynthesis in dark due to inadequate sunlight. Callus remained proliferated in dark on CCIM for 3 months at same temperature and photoperiod, and turned brownish in color. Healthy growth of callus was observed which subsequently induced the emergence of embryos from somatic calli. Healthy and well grown calli had the highest ability to survive and proliferate in darkness. The healthy (fully grown, coarse) calli were found to regenerate to embryos earlier than their weaker (lesser grown, misty) counterparts. This difference was due to the optimal density of cells within calli reported for successful embryogenesis [63, 64].

Hypocotyls induced callus, which differed in their appearance for all vector types (pKGW-RR, pHSE-401-Rep, pG-Rec and pCas-Rec), were found to be positively transformed. Calli developed from pKGW-RR transformation appeared pinkish due to the presence of DS-Red. Calli developed from pHSE-401 and pCas-Rec appeared brownish, while those developed from pG-Rec appeared as pale yellowish to greenish in color. Calli, from all transformation events, continued multiplying to yield a mass of callus covering the entire petri plate. Positively transformed somatic calli helped develop embryos.

Cotton hypocotyls were *Agro*-transformed with pKGW-RR and sampled at different stages (hypocotyls, callus induction and embryogenesis) to check the expression of DS-Red phenotypically as well as under red light (Stereomicroscope, Leica filter set DsRed for MZ F/FA: Art No. 10 447 412). As callus appeared on both sides of hypocotyls, calli from each side were carefully separated and were marked as a single transformation event. DS-Red expression from each independent event was observed carefully. Most of the calli appeared as pinkish red. Intensity of redness varied in different calli originating from different transformation events. Pink color of somatic calli was observed throughout the growth stage of callus; detaching the callus from hypocotyls till the emergence of embryos. The pKGW-RR transformed somatic calli were visibly red. The embryos within coarse reddish callus in DS-Red transformed hypocotyls (being grown in dark) appeared as small and smooth whitish beads. Embryogenic calli were not visibly red but rather fluoresce to red under stereomicroscope (Fig 4). *Agro*-DS-RED transformed embryos expressed false, pale red color when seen under microscope; it was different from the bright red color fluorescence under red light. The pKGW-RR transformation event was simultaneously confirmed visually by the red coloration of somatic and embryogenic calli due to the presence of DS-Red. Transgenic embryos were selected on the basis of developing tissues from embryos which also fluoresce when seen under red light of stereomicroscope. These embryos were carried further for inducing the development of roots and shoots.

Hypocotyls transformed with pG-Rec, pHSE-401-Rep and pCas-Rec induced calli of pale yellowish to greenish color. Selection criteria for independent transformation event remained the same, as was done for the selection of DS-Red transformed callus. That is, calli emerging

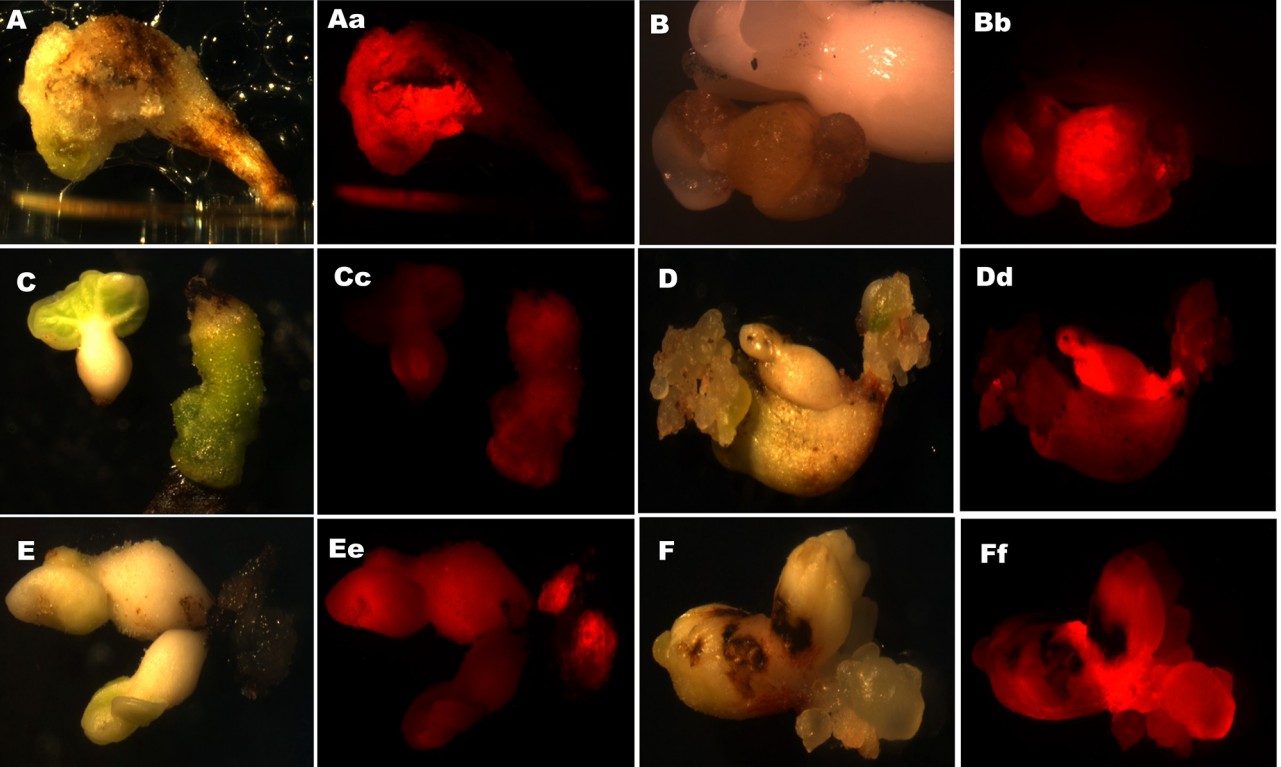

**Fig 4. Cotton embryos comparable under white light and DS-Red.** (Portion of embryos expressing strong DS-Red fluoresce to red color under stereomicroscope. Fluorescent embryos growing from highly fluorescent calli were considered as true transgenic embryos (D, E and F). DS-Red (transgene) was observed transferring from callus to embryos and ultimately transferred to roots and shoots).

from a single side of each hypocotyl was marked as an independent transformation event. Embryos originated from these independent calli were smooth whitish in appearance. Shifting embryos from dark to light helped them turn green, which seemed to initiate the process of photosynthesis. Embryos appeared as shiny white beads. Fresh plates were kept under observation for a photoperiod of 16/8 hours to induce embryos' growth, development and photosynthesis. Embryos elongated, changed the shape from globular to torpedo and developed green leaflets. When embryos developed the first two leaflets and attained the length of 10-12cm, they were observed on CRIM with no growth regulators, hormones or amino acids, but with added respective selection agent. Embryos were placed on the CRIM-I in jars. They developed roots when kept for two months on the medium of same formulation with same temperature and photoperiod conditions.

One of the major limiting steps observed was the regeneration of embryos from somatic calli. Somatic calli took 3–4 months for developing embryos on CCIM-II. Embryos being smooth white beads were clearly distinguished from callus. These beads were initially globular which became torpedo and heart shaped embryos after 2 weeks. Embryos were not visibly red unlike DS-Red transformed somatic calli but they fluoresced to red under stereomicroscope. DS-Red-transformed hypocotyls displayed healthy growth on kanamycin culture medium. Visual endorsement of transformation at each step confirmed the validity of the transformation event. Fluorescence of somatic and embryonic calli as red under the stereomicroscope also inveterated positive transformation. Different stages of embryo development showed strong localization of DS-Red within embryos (Fig 4).

Most of the selected calli showed amplification by PCR of the NptII gene in pKGW-RR and Rec cassette in pG-Rec, releasing positive fragments of 557 bp (Fig 5A) using NptII gene oligos (Table 1). The presence of the NptII gene was further confirmed by the Vir gene of *Agrobacterium* to differentiate the inherent gene of bacteria or as the whole T-DNA integrated into plant genome. The pKGW-RR DS-Red gene amplification of 2489 bp was detected using SbfI-At-DS-Red For. and NsiI-At-DS-Red Rev. oligos (Fig 5C). The Cas9 cassette was amplified as 7019 bp with M13 primers (Fig 5D). Validation of transformed pHSE-401-Rep specified the integration of Cas9 gene within positive transgenic cotton plants. Further substantiation of Cas9 gene arose from 4266 bp Cas9 gene amplification by Cas9-specific primers. The Rec cassette released 1493 and 1449 bp fragments of pG-Rec and pCas-Rec vectors respectively. Rec primers were used for amplification of Rec cassette of pG-Rec and In-Fusion primers were used for Rec cassette amplification of pCas-Rec. The mean transformation frequency (the number of positively transformed plants divided by total number of hypocotyls infected by *Agrobacterium*) was ~ 3–4% transgenic cotton plants.

Individual transgenic event was separated carefully by detaching callus from either side of each hypocotyl. We transformed nearly 10,000 hypocotyls with each plasmid. Nearly 40 independent transformation events were selected for carrying further to the regeneration stage. Calli were independently selected on the basis of antibiotic resistance genes and the transgene analysis. A range of 19–23 embryos was observed from each independent transformation event. Approximately, a sum of 17 positively transformed plantlets (pKGW-RR, pG-Rec, pHSE-401–Rep and pCas-Rec) were able to grow till maturity (Table 2). DS-Red-transformed hypocotyls showed red coloration on both cut ends. Calli from both cut ends were isolated to be identified as separate transformation events. Healthy calli separated from hypocotyls appeared as red when selected on kanamycin plates. Red coloration retained throughout the callus growth stage till the development of embryos. Some of the kanamycin selected calli did

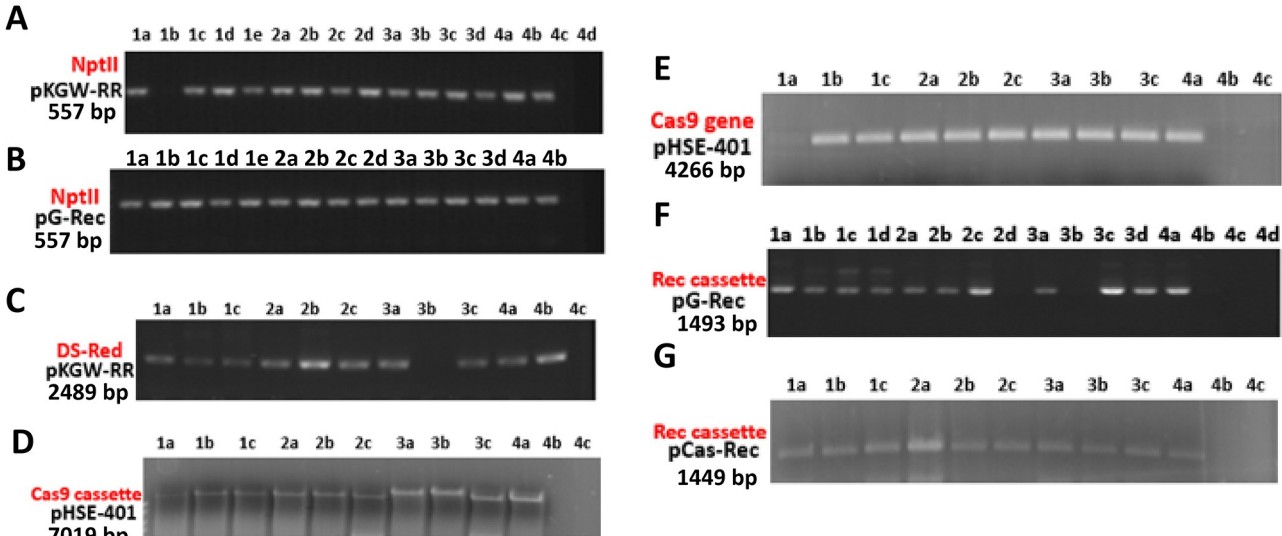

**Fig 5. PCR amplification of pKGW-RR, pG-Rec, pHSE-401-Rep and pCas9-Rec.** (5A) Kanamycin gene amplification (557 bp) in DNA samples transformed with DS-Red vector. (5B) Kanamycin gene amplification (557 bp) in DNA samples transformed with pG-Rec vector. (5C) Complete DS-Red cassette amplification (2489 bp) in DNA samples putatively transformed with pKGW-RR vector. (5D) Whole Cas9 cassette amplification (7019 bp) in plants transformed with pHSE-401-Rep. (5E) Cas9 gene amplification (4266 bp) in plants transformed with pHSE-401-Rep. (5F) Whole Rec cassette amplification in plants transformed with pG-Rec (1493 bp). (5G) Whole Rec cassette amplification (1449 bp) in plants transformed with pCas-Rec vector.

**Table 2. Transformation events.** Number of independent transformation events, regenerated embryos, positively transformed plantlets and transformation percentage per batch measured for four plasmids (1, 2, 3 and 4 for pKGW-RR, pG-Rec, pHSE-401–Rep and pCas-Rec respectively).

| Serial No. | Batch No. | Transformation Event No. | No. of transgenic plants | Transformation percentage | Transgene Integrated | Binary vector | Reference |
|---|---|---|---|---|---|---|---|
| 1 | I | 01001001–076 | 17 | 37.78 | DS-RED | pKGW-RR | [65] |
| 2 | | 02001001–081 | 14 | 31.11 | Cas9-gRNA | pHSE-401-Rep | Addgene |
| 3 | | 03001001–059 | 06 | 13.33 | Rec | pG-Rec | [62] |
| 4 | | 04001001–063 | 08 | 17.78 | Rec-Cas9 | pCas-Rec | This study |
| 5 | II | 01002001–041 | 12 | 44.44 | DS-RED | pKGW-RR | [65] |
| 6 | | 02002001–033 | 7 | 25.93 | Cas9-gRNA | pHSE-401-Rep | Addgene |
| 7 | | 03002001–016 | 3 | 11.11 | Rec | pG-Rec | [62] |
| 8 | | 04002001–025 | 5 | 18.52 | Rec-Cas9 | pCas-Rec | This study |
| 9 | III | 01003001–004 | 5 | 41.67 | DS-RED | pKGW-RR | [65] |
| 10 | | 02003001–008 | 7 | 58.33 | Cas9-gRNA | pHSE-401-Rep | Addgene |
| 11 | | 03003001–007 | 0 | 0.00 | Rec | pG-Rec | [62] |
| 12 | | 04003001–005 | 0 | 0.00 | Rec-Cas9 | pCas-Rec | This study |
| 13 | IV | 01004001–004 | 11 | 61.11 | DS-RED | pKGW-RR | [65] |
| 14 | | 02004001004 | 3 | 16.67 | Cas9-gRNA | pHSE-401-Rep | Addgene |
| 15 | | 03004001004 | 0 | 0.00 | Rec | pG-Rec | [62] |
| 16 | | 04004001004 | 4 | 22.22 | Rec-Cas9 | pCas-Rec | This study |

not show red coloration. It could be because of the marker gene silencing while integrating into genome or due to chimeric nature of transformed hypocotyls. Due to regeneration of cotton via embryogenesis, it is reported that even the chimeric calli were considered as transgene positive and express foreign gene [53].

PCR positive transgenic lines were subjected to DNA blot hybridization analysis for the determination of transgene copy number in cotton genome. Genomic DNA digested with restriction enzyme carried a single restriction site in T-DNA. For Southern blot analysis, hybridization was done with gene probe of NptII selectable marker to determine the number of integrated gene copies from the number of hybridized fragments. The copy number measurements from Rec, Rep and Cas-9 genes are provided in Fig 6.

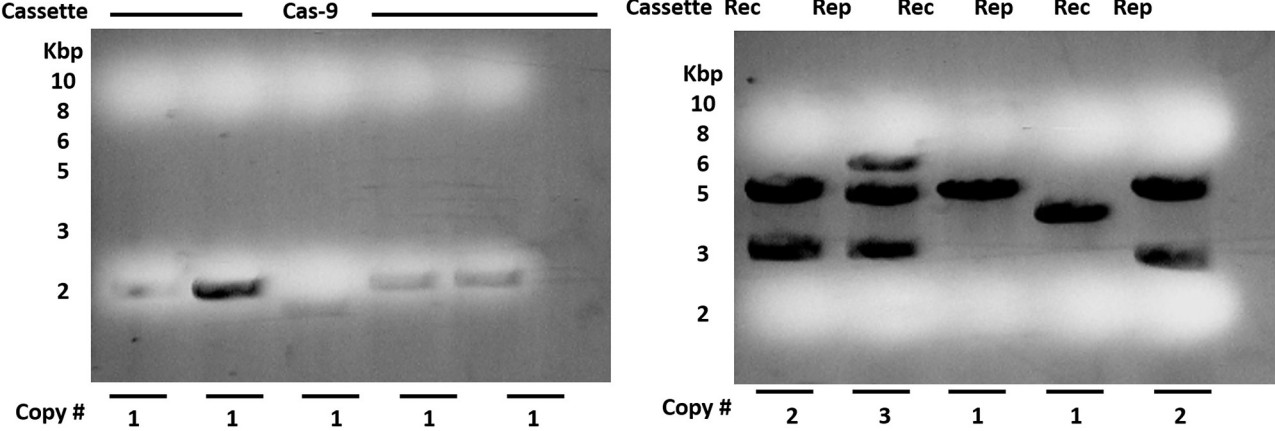

**Fig 6. Copy number measurement in cotton transgenics.** (Blot hybridization analysis in transgenic cotton genomic DNA hybridized with labeled probe (nptII, hptII and Cas9 genes). The approximate DNA length is shown to the right and cassettes integrated in particular transgenic are shown above the autoradiograph).

## Discussion

*Agrobacterium*-mediated transformation is able to introduce foreign DNA into the genome of most dicot species [66]. Developing hypocotyls from mature cotton seeds in soil were infected by *Agrobacterium* harboring four distinct constructs. T-DNA having integrated cassette translocated through the cut ends of hypocotyls to get incorporated into the genome. Somatic and embryonic calli were selected in a non-lethal pattern using DS-Red fluorescence. Putative transformed embryos were visually selected from non-transformed cells. Antibiotic selection and transgene analysis further eased the screening of transgenic plants from non-transgenic or putative ones.

DS-Red was found to be a strong visual marker for identifying positive transformation and regeneration. Previously GFP has been reported for monitoring transformation and gene expression in cotton [53, 67]. Visualization of cotton transformation by DS-Red helped develop transgenic cotton lines with three other plasmids (pG-Rec, pHSE-401-Rep and pCas-Rec).

Three plasmids (pG-Rec, pHSE-401-Rep and pCas-Rec) were individually transformed to be used as a basal step for three independent projects. The pG-Rec vector was used initially to develop base plants of cotton for recombinase mediated gene stacking. This vector integrated PhiC31 mediated attP sites and Cre mediated LoxP sites within Coker-312 genome. These sites were identified as basal sites for specifying locus of first incoming exchange plasmid. LoxP sites integrated within Coker-312 plants will recombine with LoxP sites within incoming exchange plasmid for the removal of marker gene integrated during transformation of pG-Rec. First incoming exchange vector will bring Bxb1 mediated attB sites to recombine with phiC31 mediated attP sites integrated through the base vector. PhiC31 mediated integration will result in addition of first gene of interest (GOI) and Cre mediated marker gene removal within a single step. First incoming exchange vector will also bring additional attB sites to serve as the recombination sites for future incoming exchange vector to be transformed. The current study of transforming pG-Rec initiated the project of gene stacking in cotton.

pHSE-401-Rep was used to integrate Cas9 gene into the cotton genome for testing CRISPR/Cas system affectivity against Cotton Leaf Curl Virus (CLCuV) infectious clone. Primarily, the CRISPR system was tested against cotton leaf curl infectious clone infectivity in *Nicotiana banthamiana* to test its efficacy [68]. This vector carried a hygromycin gene, which established the transformation reliability against other selective agents. pHSE-401-Rep was transformed for initiating the project of controlling CLCuV, as this vector carried gRNA against Rep conserved region of CLCuV.

pCas-Rec was transformed as a third independent vector for developing Cas9 based cotton plants. This vector carried a Cas9 gene and a Rec cassette (carrying PhiC31 and Bxb1 recombinase mediated DNA recognition sites) for specifying Cas9 gene locus. Site-specific Cas9 base plants will allow a range of gRNAs to be inserted either transiently or stably. gRNAs can also be transformed stably by recombinase mediated integration event.

To conclude, we accomplished these basal experiments to initiate multiple branches of cotton transgenics. We established pG-Rec transformed cotton founder plants to start a gene stacking project of cotton. A Cas9 gene with gRNAs against conserved replication region of CLCuV was piled up with AVP1 gene [69] for augmenting the plant growth and survival of cotton bolls. These founder plants will be grown further for developing Rec based cotton founder lines. pHSE-401-Rep transformed cotton founder plants will be used to target CLCuV to combat the disease and to minimize the loss associated with CLCuD. Moreover, Cas9-Rep founder lines will also be initiated to be feasible for multiple gRNAs against

numerous DNA sequences. pCas-Rec transformed founder plants were produced to develop a site specific Cas9 originator line of cotton. By developing a stable Cas9 founder line, gRNA can be introduced by transient transformation to target the desired traits. Cas9 gene will be residing in cotton's genome and gRNAs will transiently work to knock out the lethal genes (Fig 7).

Cotton embryogenesis efficiency has been improved by many researchers but numerous obstacles are still on their way to increase efficiency of cotton transformation and somatic embryogenesis. Firstly, it requires numerous subcultures that leads to the development of somaclonal variations and longer time for tissue culture. Secondly, the regeneration capacity of plant is low and highly genotype dependent. Only a few genotypes among 50 species of cotton have been induced to proceed for plant regeneration and somatic embryogenesis. An important factor involved in the success of the development of cotton somatic embryogenesis is desiccation stress. Desiccation and metabolic stress might enhance somatic embryogenesis, maturation of somatic embryos and plant recovery on shifting them to soil [70].

CRISPR/Cas9 genome editing system being highly efficient and flexible finds its many applications in cotton. As *Gossypium hirsutum* Chloroplast alterados (GHCLA-1) *GH* Vacular H$^+$ pyrophosphatse gene (GHVP) were targeted by CRISPR for gene insertion/substitution and deletion respectively [31]. Some of the widespread CRISPR applications of cotton genome editing includes targeted knock out of GFP gene [71], multilocus genome editing in cotton allotetraploid species [72] and formation of enhanced lateral roots by CRISPR in cotton [72]. Latest advancements in genome sequencing and editing technologies enabled scientists to improve cotton. Targeted genes for genome editing are more exposed due to multiomics and whole genome sequencing. Combined applications of both technologies have opened new horizons for improving agronomic traits and for the production of pharmaceuticals and biofuel from cotton. This will enable cotton farmers to increase their income, improve cotton quality and enhance its competitiveness against synthetic fiber across the globe. A fast application of genome sequencing and editing techniques is required to be integrated into precision breeding for not only cotton but for all the crops. Thereby, exploring latest strategies will increase the applications of these technologies for the improvement of crops [73] (Fig 8).

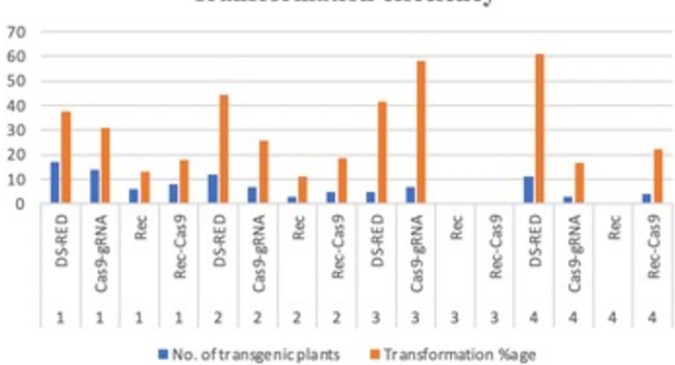

**Fig 7. Transformation effeciency: Transformation effeciency depicted in four distinct batches of all the four plasmids transformed (pKGWRR, pHSE-401, pG-Rec and pCas-Rec).** Transformation percentage per batch is calculated on the basis of positively transformed plants.

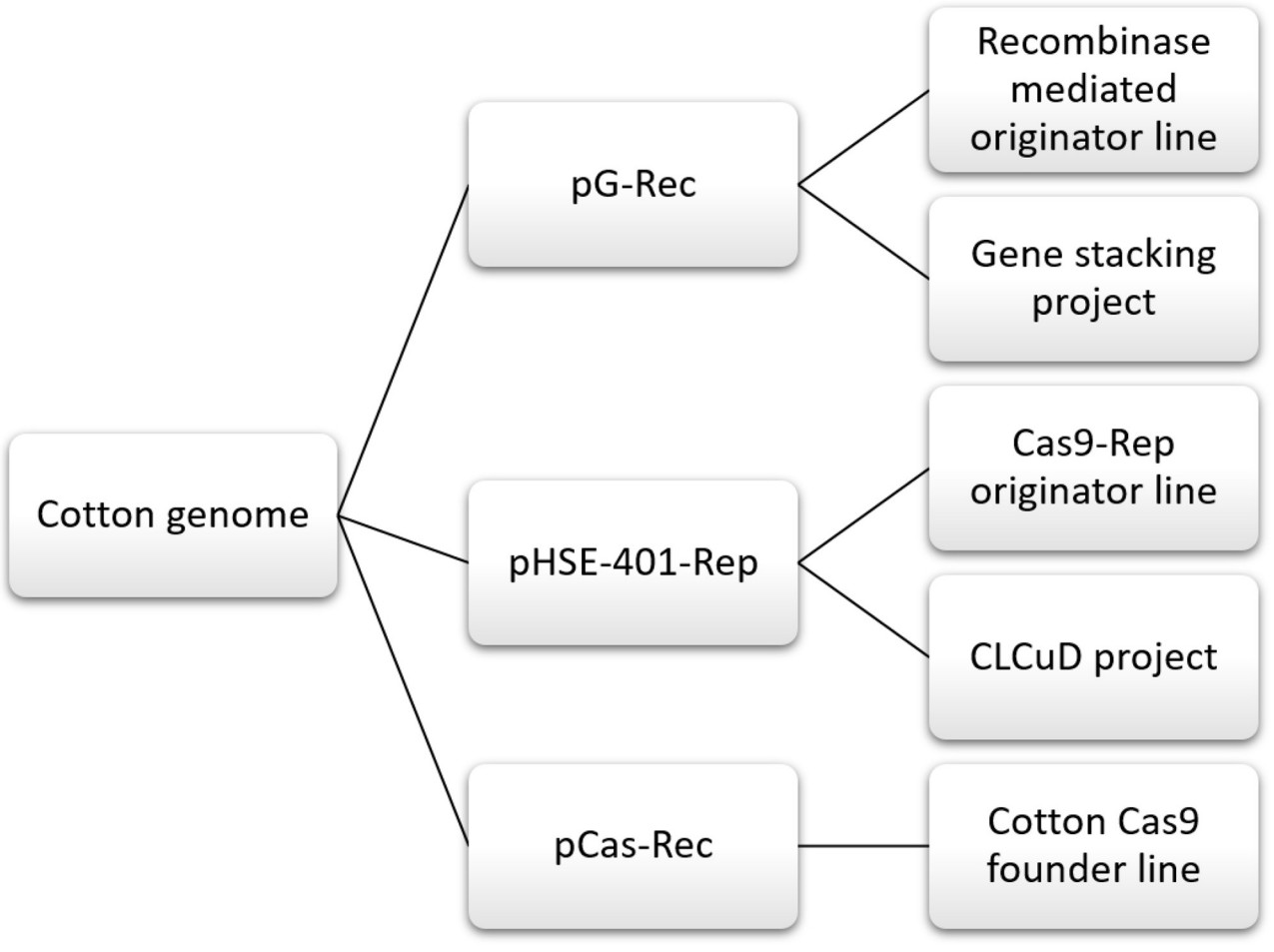

**Fig 8. Way forward to cotton projects.**

## Supporting information

**S1 Raw images.**
(PDF)

## Acknowledgments

The whole research work was done by Sabin Aslam. Sultan Habibullah Khan did conceptualization. Sriema Lalani Walawage and Aftab Ahmad guided about methodology. Abhaya M. Dandekar did validation and visualization of results. We are thankful to Dr. James Thomson for his kind guidance about lining up of gene stacking projects. We are also thankful to Dr. Mohsin Raza for statistical analysis.

## Author Contributions

**Conceptualization:** Sultan Habibullah Khan.

**Methodology:** Sabin Aslam, Aftab Ahmad, Sriema Lalani Walawage.

**Visualization:** Abhaya M. Dandekar.

**Writing – original draft:** Sabin Aslam.

**Writing – review & editing:** Sabin Aslam.

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
