## [Decision Letter · Decision Letter 0]

2 Nov 2020

PONE-D-20-28782

Founder transformants of cotton (*Gossypium hirsutum* L.) obtained through the introduction of DS-Red, Rec, Rep and CRISPR/Cas9 expressing constructs for developing base lines of recombinase mediated gene stacking

PLOS ONE

Dear Dr. ASLAM,

Thank you for submitting your manuscript to PLOS ONE. After careful consideration, we feel that it has merit but does not fully meet PLOS ONE’s publication criteria as it currently stands. Therefore, we invite you to submit a revised version of the manuscript that addresses the points raised during the review process.

We look forward to receiving your revised manuscript.

Kind regards,

Allah Bakhsh

Academic Editor

PLOS ONE

Journal Requirements:

Reviewers' comments:

Reviewer's Responses to Questions

**Comments to the Author**

1. Is the manuscript technically sound, and do the data support the conclusions?

Reviewer #1: No

Reviewer #2: Yes

Reviewer #3: Yes

2. Has the statistical analysis been performed appropriately and rigorously? 

Reviewer #1: N/A

Reviewer #2: N/A

Reviewer #3: N/A

3. Have the authors made all data underlying the findings in their manuscript fully available?

Reviewer #1: Yes

Reviewer #2: No

Reviewer #3: Yes

4. Is the manuscript presented in an intelligible fashion and written in standard English?

Reviewer #1: No

Reviewer #2: Yes

Reviewer #3: Yes

5. Review Comments to the Author

Reviewer #1: In the paper by Aslam et al., 2020, authors claim the development of founder lines of Ds-Red, Rec, Rep and CRISPR/Cas9 that can be used for recombinase mediated gene stacking. However, the study has many flaws, the paper is poorly written and not properly organized before submission. The manuscript has figures and data that only describe the cotton transformation steps. I have major concerns about the novelty of the manuscript:

- The methods section is unnecessarily lengthy. Are these methods of cotton transformation and southern blotting developed by the authors, if yes, then highlight the parts which are developed in this study and minimize rest of the portions with proper references. And if not, which is most likely, then revise and summarize the methods with references.

- The Results section is real disaster. Most of the parts of results revolving around the methodology of tissue culture with no real scientific findings. The headings in the results sections are written as figures titles: line 353-357, line 396, line 423. Similarly in the discussion, line 477. It seems authors did not read carefully before submission.

- Authors assumed in the line 465-476, how these lines will be helpful for future studies. However, they must show some figures and data to prove their claims.

Reviewer #2: In this manuscript, the authors developed a several resources for obtaining transgenic and genome editing cotton, it is an interested and useful research. The experimental design is good and will enhance the related cotton transgenic and genome editing in the future.

1. Page 2, line 45. “This virus was originated in Nigeria in 1912”, I am sure it was reported in 1912. Does it mean “it was originated in 1912”? It is better to change it.

2. It is true that it is hard to obtain regenerated plants from cotton tissue culture, however, a report in 2009 [Highly efficient plant regeneration through somatic embryogenesis in 20 elite commercial cotton (Gossypium hirsutum L.) cultivars. Plant Omics, 2009, 2(6): 259-268], described a method for obtaining regenerated from almost all cotton cultivars, it may be better to discuss this paper in this manuscript.

3. Page 8. Line 162, “Coker-312 seeds were germinated at 28oC in darkness for two weeks in sunshine mix soil”, is it right, in my observation, in the darkness under 28C, the cotton plant can reach the sizes in the figures in about 7-10 days.

4. When do infection with Agrobacterium, what the concentrations for the agrobacteria?

5. “4.33 g Murashige and Skoog minimal organics medium, 1 mL/L MS vitamin stock” should be called regular MS medium, it does not need to list it out each time.

6. This manuscript dinot organized very well, they should organized as: 1. Introduction, 2. Methods and Materials (following subtitle, 2,1, 2,2,---). 3. Results, 4. Discussions.

7. One of the major goals of this manuscript is to develop a baseline for CRISPR/Cas genome editing, it is better to summarize the application of CRISPR in cotton.

8. Recently, a great paper published in Trends in Biotechnology (From Sequencing to Genome Editing for Cotton Improvement, https://doi.org/10.1016/j.tibtech.2020.09.001), discuss the similar issues discussed in this manuscript. I suggest the author read this paper and discuss the issue and strategies risen in this paper.

9. Figure 6, what does it mean here for cop number? I guess this for an individual gene not a coup number? How they would detect copy number based on southern blotting?

Reviewer #3: The presented article is quite original and contains the results of important laboratory studies. Transgenic cotton lines obtained can be used in combating CLCuD. By developing a stable Cas9 founder line, gRNA can be introduced by transient

transformation to target desired trait. Cas9 gene will be sitting inside the genome of cotton and gRNAs can transiently work to knock out the lethal genes.

Before the article can be accepted for publication, the following minor changes must be made.

-Please do not use two parentheses together. You can give the descriptions in the same brackets using semicolon (;)

-Page 3 Line 65: Couldn't can be Could not

-Page 4 Line 69: Better to be "For the eradication of disease completely,"

- Southern Blot analysis method can be shortened in method.

- Figure captions should be more descriptive and contain a full title.

-The quality of all figures should be increased. Better regeneration pictures should be given.

6. PLOS authors have the option to publish the peer review history of their article (what does this mean?). If published, this will include your full peer review and any attached files.

Reviewer #1: **Yes: **Haroon Butt

Reviewer #2: No

Reviewer #3: No

---

## [Author Response · Author response to Decision Letter 0]

10 Mar 2021

Thank you very much for the detailed suggestions to help improve the article.

---

## [Decision Letter · Decision Letter 1]

6 Apr 2021

PONE-D-20-28782R1

Founder transformants of cotton (*Gossypium hirsutum* L.) obtained through the introduction of DS-Red, Rec, Rep and CRISPR/Cas9 expressing constructs for developing base lines of recombinase mediated gene stacking

PLOS ONE

Dear Dr. ASLAM,

Thank you for submitting your manuscript to PLOS ONE. After careful consideration, we feel that it has merit but does not fully meet PLOS ONE’s publication criteria as it currently stands. Therefore, we invite you to submit a revised version of the manuscript that addresses the points raised during the review process.

Dear authors, I have received comments from the reviewers on revised version of MS. By agreeing with reviewers, I would recommend you to address all comments properly and to address redundancy in MS. I will be looking ahead to see the revised version in short time.

We look forward to receiving your revised manuscript.

Kind regards,

Allah Bakhsh

Academic Editor

PLOS ONE

Reviewers' comments:

Reviewer's Responses to Questions

**Comments to the Author**

1. If the authors have adequately addressed your comments raised in a previous round of review and you feel that this manuscript is now acceptable for publication, you may indicate that here to bypass the “Comments to the Author” section, enter your conflict of interest statement in the “Confidential to Editor” section, and submit your "Accept" recommendation.

Reviewer #1: (No Response)

Reviewer #2: (No Response)

2. Is the manuscript technically sound, and do the data support the conclusions?

Reviewer #1: No

Reviewer #2: Yes

3. Has the statistical analysis been performed appropriately and rigorously? 

Reviewer #1: No

Reviewer #2: No

4. Have the authors made all data underlying the findings in their manuscript fully available?

Reviewer #1: Yes

Reviewer #2: Yes

5. Is the manuscript presented in an intelligible fashion and written in standard English?

Reviewer #1: No

Reviewer #2: No

6. Review Comments to the Author

Reviewer #1: The authors did not address my previous comments.

- The methods section is unnecessarily lengthy. Are these methods of cotton transformation and southern blotting developed by the authors, if yes, then highlight the parts which are developed in this study and minimize rest of the portions with proper references. And if not, which is most likely, then revise and summarize the methods with references.

- The Results section is real disaster. Most of the parts of results revolving around the methodology of tissue culture with no real scientific findings. The headings in the results sections are written as figures titles. Write the results headings different than the figure titles.

Reviewer #2: The authors have addressed some comments but many not, at the same time they also generated several new big issues on this revised manuscript.

They added some research on CRISPR/Cas9 genome editing in cotton. It is very surprised, they cited almost all related literatures except the one that employing CRISPR/Cas9 genome editing technology in cotton for the first time successfully knocking out a cotton gene (Li, C et al. A high-efficiency CRISPR/Cas9 system for targeted mutagenesis in Cotton (Gossypium hirsutum L.). Sci Rep 7, 43902 (2017). https://doi.org/10.1038/srep43902), what reason they did not cite this paper?

Many methods are still not clear, such as when they infect cotton explants, how they prepare the Agrobacteria and what concentration of Agrobacterium they used? How they performed the air-dried for 2-3 min in LFC. All these method affected their results.

Still unclear how they made the MS medium, 1mL/L MS vitamin stock, what is the concentrations?

Fig 6, it is hard to understand, they should explain more.

7. PLOS authors have the option to publish the peer review history of their article (what does this mean?). If published, this will include your full peer review and any attached files.

Reviewer #1: No

Reviewer #2: No

---

## [Author Response · Author response to Decision Letter 1]

3 Jun 2021

I have made all changes according to reviewers comments and modified the manuscript extensively.

---

## [Decision Letter · Decision Letter 2]

21 Jul 2021

PONE-D-20-28782R2

Founder transformants of cotton (*Gossypium hirsutum* L.) obtained through the introduction of DS-Red, Rec, Rep and CRISPR/Cas9 expressing constructs for developing base lines of recombinase mediated gene stacking

PLOS ONE

Dear Dr. ASLAM,

Thank you for submitting your manuscript to PLOS ONE. After careful consideration, we feel that it has merit but does not fully meet PLOS ONE’s publication criteria as it currently stands. Therefore, we invite you to submit a revised version of the manuscript that addresses the points raised during the review process.

Dear authors, I have now received comments on your revised version from both reviewers. You will see that they are asking to focus the point raised by them. Please write a good rebuttal letter keeping in view the comments from both reviewers. Please returned the revised version once the raised points have been addressed well.

We look forward to receiving your revised manuscript.

Kind regards,

Allah Bakhsh

Academic Editor

PLOS ONE

Reviewers' comments:

Reviewer's Responses to Questions

**Comments to the Author**

1. If the authors have adequately addressed your comments raised in a previous round of review and you feel that this manuscript is now acceptable for publication, you may indicate that here to bypass the “Comments to the Author” section, enter your conflict of interest statement in the “Confidential to Editor” section, and submit your "Accept" recommendation.

Reviewer #1: (No Response)

Reviewer #2: (No Response)

2. Is the manuscript technically sound, and do the data support the conclusions?

Reviewer #1: Partly

Reviewer #2: Partly

3. Has the statistical analysis been performed appropriately and rigorously? 

Reviewer #1: N/A

Reviewer #2: No

4. Have the authors made all data underlying the findings in their manuscript fully available?

Reviewer #1: Yes

Reviewer #2: (No Response)

5. Is the manuscript presented in an intelligible fashion and written in standard English?

Reviewer #1: No

Reviewer #2: No

6. Review Comments to the Author

Reviewer #1: The comments are not addressed seriously in the revisions. All the sections of paper are below the quality of PLOS ONE.

Authors fail to show figures and data to prove their claims.

Reviewer #2: The authors did not take serious revision as the first revision. Although the author did some revision, many issues have not been addressed. They should also list a point to point response to each comment risen from the reviewers. It is hard to understand which comment they addressed and which ones not. How they addressed in the manuscript?

I suggest the authors go back the first two round reviews, and re-read the comments from all reviewers and addressed one by one and give a detailed response.

7. PLOS authors have the option to publish the peer review history of their article (what does this mean?). If published, this will include your full peer review and any attached files.

Reviewer #1: No

Reviewer #2: No

---

## [Author Response · Author response to Decision Letter 2]

1 Sep 2021

I have submitted a detailed response to reviewers letter. Please proceed for publication phase.

---

## [Decision Letter · Decision Letter 3]

18 Jan 2022

Founder transformants of cotton (*Gossypium hirsutum* L.) obtained through the introduction of DS-Red, Rec, Rep and CRISPR/Cas9 expressing constructs for developing base lines of recombinase mediated gene stacking

PONE-D-20-28782R3

Dear Dr. ASLAM,

We’re pleased to inform you that your manuscript has been judged scientifically suitable for publication and will be formally accepted for publication once it meets all outstanding technical requirements.

Kind regards,

Allah Bakhsh

Academic Editor

PLOS ONE

Additional Editor Comments (optional):

Reviewers' comments:

Reviewer's Responses to Questions

**Comments to the Author**

1. If the authors have adequately addressed your comments raised in a previous round of review and you feel that this manuscript is now acceptable for publication, you may indicate that here to bypass the “Comments to the Author” section, enter your conflict of interest statement in the “Confidential to Editor” section, and submit your "Accept" recommendation.

Reviewer #4: All comments have been addressed

2. Is the manuscript technically sound, and do the data support the conclusions?

Reviewer #4: Yes

3. Has the statistical analysis been performed appropriately and rigorously? 

Reviewer #4: Yes

4. Have the authors made all data underlying the findings in their manuscript fully available?

Reviewer #4: Yes

5. Is the manuscript presented in an intelligible fashion and written in standard English?

Reviewer #4: Yes

6. Review Comments to the Author

Reviewer #4: I thank the authors for correcting the MS according the reviewer's comments. Now it is more easier to understand the MS.

7. PLOS authors have the option to publish the peer review history of their article (what does this mean?). If published, this will include your full peer review and any attached files.

Reviewer #4: No

---

## [Editor Report · Acceptance letter]

21 Jan 2022

PONE-D-20-28782R3 

Founder transformants of cotton (*Gossypium hirsutum* L.) obtained through the introduction of DS-Red, Rec, Rep and CRISPR/Cas9 expressing constructs for developing base lines of recombinase mediated gene stacking 

Dear Dr. Aslam:

I'm pleased to inform you that your manuscript has been deemed suitable for publication in PLOS ONE. Congratulations! Your manuscript is now with our production department. 

Kind regards, 

on behalf of

Dr. Allah Bakhsh 

Academic Editor

PLOS ONE